# Real-time measurement of radionuclide concentrations and its impact on inverse modeling of $^{106}$Ru release in the fall of 2017

Ondřej Tichý[1], Miroslav Hýža[2], Nikolaos Evangeliou[3], and Václav Šmídl[1]

[1]The Czech Academy of Sciences, Institute of Information Theory and Automation, Prague, Czech Republic
[2]National Radiation Protection Institute, Prague, Czech Republic
[3]Norwegian Institute for Air Research (NILU), Kjeller, Norway

**Correspondence:** Ondřej Tichý (otichy@utia.cas.cz)

**Abstract.** Low concentrations of $^{106}$Ru were detected across Europe at the turn of September and October 2017. The origin of $^{106}$Ru has still not been confirmed; however, current studies agree that the release occurred probably near Mayak in the southern Urals. The source reconstructions are mostly based on an analysis of concentration measurements coupled with an atmospheric transport model. Since reasonable temporal resolution of concentration measurements is crucial for proper source term reconstruction, the standard one-week sampling interval could be limiting. In this paper, we present an investigation of the usability of the newly developed AMARA and CEGAM real-time monitoring systems, which are based on the gamma-ray counting of aerosol filters and allow to determine the moment when $^{106}$Ru arrived at the monitoring site within approx. one hour and activity concentrations as low as several mBq/m$^3$ can be detected in 4-hour intervals. These high-resolution data were used for inverse modeling of the $^{106}$Ru release. We perform backward runs of the Hysplit atmospheric transport model driven with meteorological data from the global forecast system (GFS) and we construct a source-receptor sensitivity (SRS) matrix for each grid cell of our domain. Then, we use our least-squares with adaptive prior covariance (LS-APC) method to estimate possible locations of the release and the source term of the release. On Czech monitoring data, the use of concentration measurements from the standard regime and from the real-time regime is compared and better source reconstruction for the real-time data is demonstrated in the sense of the location of the source and also the temporal resolution of the source. The estimated release location, Mayak, and the total estimated source term, $237 \pm 107$ TBq, are in agreement with previous studies. Finally, the results based on the Czech monitoring data are validated with the IAEA reported dataset with a much better spatial resolution, and the agreement between the IAEA dataset and our reconstruction is demonstrated. In addition, we validated our findings also using the FLEXPART model coupled with meteorological analyses from the European Centre for Medium-Range Weather Forecasts (ECMWF).

## 1 Introduction

At the turn of September and October 2017, low concentrations of $^{106}$Ru of unknown origin were detected in the atmosphere in the Czech Republic. Immediate communication with other European laboratories involved in the RO5 (Ring of 5) network (Masson et al., 2011) confirmed that this was a Europe-wide occurrence. Although the concentration was low (tens of mBq/m$^3$) and was of no health risk, the unknown origin of $^{106}$Ru raised concerns. Therefore, very shortly after the first detections, efforts

were made to estimate the source location based on the RO5 data. Initial analyses pointed to a possible source located to the east of the Czech Republic. As the dataset grew, this estimate was refined to the Urals region as the most probable location (Kovalets and Romanenko, 2017). The released $^{106}$Ru activity was estimated to be several hundred TBq (Saunier et al., 2019; Western et al., 2020).

Since $^{106}$Ru is a fission product produced in a nuclear reactor, the question arose about the nature of the source. A nuclear reactor accident was rejected because, in this case, other radionuclides would have been detected besides $^{106}$Ru, similarly as during the Chernobyl NPP accident (UNSCEAR, 2000). For example, during post-Chernobyl monitoring, the detected $^{106}$Ru was by 2 or 3 orders of magnitude higher and was accompanied by a complex mix of radionuclides, including $^{131}$I, $^{132}$Te, $^{137}$Cs, $^{134}$Cs, $^{140}$La, and $^{103}$Ru (CHZ, 1987).

Other working hypotheses included the melting of a radioisotope thermoelectric generator (RTG) or of a medical source, since $^{106}$Ru is used in medicine for the treatment of ophthalmic tumors (Takiar et al., 2015). In several samples where the $^{106}$Ru activity was relatively high, we also detected $^{103}$Ru isotope, but at much lower concentrations. The activity ratio of $^{106}$Ru/$^{103}$Ru was approx. 4000 (after the Chernobyl accident, the ratio was approximately 0.12), which suggests that the ruthenium was extracted from relatively fresh nuclear fuel (approximately 2 years). Since medical sources and RTG would

neither explain the occurrence of Ru-106 nor the large source of several hundred TBq, fresh nuclear fuel is the most likely candidate.

In the end, an industrial source was identified as the most probable explanation – most likely a fuel reprocessing plant. This conclusion is supported by historical evidence since we have observed several such events in the past – Tomsk (Tcherkezian et al., 1995), Savannah River (Carlton and Denham, 1997) and La Hague (ACRO, 2002). Based on these reports, it can be

concluded that a selective release of $^{106}$Ru is possible during certain stages of fuel reprocessing or vitrification of fuel in the form of highly volatile $RuO_4$ which can escape into the environment even when aerosol filters are employed. $RuO_4$ then condenses in the colder air and can be further transported over long distances attached to atmospheric aerosol. There are two known plants in the southern Urals region which come into consideration – Mayak and Dimitrovgrad. Both are located within the region estimated by atmospheric transport modeling (ATM). Moreover, measurements performed by Roshydromet confirm

a positive detection of $^{106}$Ru in aerosols and in the fallout in the Chelyabinsk region (Shershakov et al., 2019).

Multiple investigations using different data sets and methodologies have now been performed with the same conclusion, indicating the Mayak plant as the probable source location (Masson et al., 2019; Saunier et al., 2019; Maffezzoli et al., 2019; De Meutter et al., 2019; Le Brazidec et al., 2020). Masson et al. (2019) presented a comprehensive event analysis, including a detailed radioruthenium forensic investigation, and speculated on the possibility of $^{106}$Ru release during the production of the

$^{144}$Ce source for the SOX-Borexino at the Gran Sasso National Laboratory (also suggested by Bossew et al. (2019)). Nonetheless, the Russian authorities deny any leakage from the Mayak plant (Nikitina and Slobodenyuk, 2018). Current estimates of $^{106}$Ru source location and source term are mainly based on an analysis of ambient measurements of $^{106}$Ru concentrations.

There is always a trade-off between sensitivity and timely reporting of concentration results, and the standard procedure provides a rather poor time resolution of the concentration monitoring data for the purposes of ATM analyses. The time delay

between the possible arrival of the contamination at the monitoring site and its detection can easily be as long as one week. Long-term shortening of the sampling interval below one day is virtually unachievable, mainly for logistic reasons.

This limitation is of great research interest at the National Radiation Protection Institute (NRPI), Czech Republic, where near real-time monitoring systems (AMARA and CEGAM, see Section 2 for a detailed description ) are currently under development. Both systems yield minimum detectable activity (MDAC) at a level of 1 mBq/m$^3$ which was sufficient to detect $^{106}$Ru during the 2017 episode. We were able to perform an experimental run of the AMARA device, and we managed to detect the exact moment when the contamination arrived. These real-time monitoring data were then used for source localization and the results were compared with the standard time resolution. For this purpose, we use a Bayesian inversion method called the least squares with an adaptive prior covariance (LS-APC) method (Tichý et al., 2016) which was later extended also for the source location problem (Tichý et al., 2017).

Our aim is to use the data from the Czech radiation monitoring network to investigate two points. First, we will study the influence of the real-time monitoring data on the resulting estimate of the temporal profile of the emission. Our hypothesis is that the use of real-time monitoring data should lead to more time-specific estimates. Second, we will investigate and discuss what information can be estimated from the Czech monitoring data only. This task is very challenging since it implies a very sparse monitoring network due to the small area of the Czech Republic in comparison with the relevant Europe-Asia spatial domain. The results will be validated and will be compared with results of the much larger IAEA dataset (IAEA, 2017).

## 2 Measurement methodology and datasets

### 2.1 Standard sampling and measurement procedure

In the Czech radiation monitoring network (RMN), aerosol samples are taken from 10 permanent monitoring sites which are equipped with high volume aerosol samplers with a flow rate in the range of 150 – 900 m$^3$/h. In addition to these monitoring sites, radionuclides are also monitored in the local networks in the vicinity of the nuclear power plants in the Czech Republic – these data are not included in the analysis.

The standard sampling frequency is usually once or twice a week. Combined weekly samples are subjected to semiconductor gamma spectrometry, with no further treatment, at four RMN laboratories. Preliminary measurement of aerosol filters starts a few hours after the end of the sampling, to allow time for the short-lived radon progenies to decay. Otherwise, they would significantly affect the measurement sensitivity. The preliminary measurements last approximately 5 hours, after which the detection limit (minimum detectable activity – MDAC) is at a level of 10 $\mu$Bq/m$^3$. Consequently, a detailed measurement lasting approx. 5 days is performed, after which the sub–$\mu$Bq/m$^3$ MDAC level is achieved.

$^{106}$Ru is a $\beta$-emitter and therefore cannot itself be detected by means of gamma-ray spectrometry. $^{106}$Ru activity is determined on the basis of its short-lived progeny $^{106}$Rh, which emits several gamma rays of convenient energy and intensity (622 keV and 1050 keV being the most prominent). In order to determine the activity accurately, it is necessary to correct for true coincidence effects, as $^{106}$Rh emits gamma photons in cascades. By failing to do this, one can easily underestimate the activity by 15–20%.

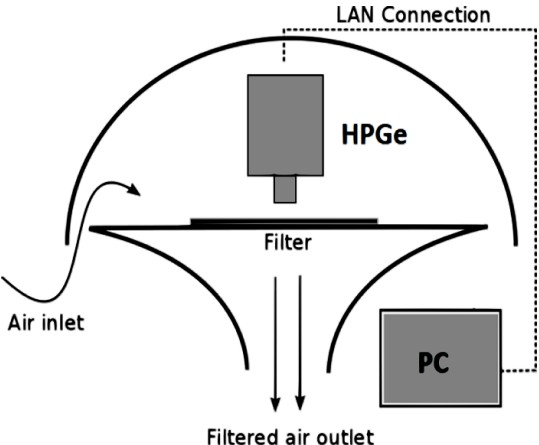

**Figure 1.** AMARA system schematics; the activity deposition is measured using an HPGe detector above an aerosol filter during sampling.

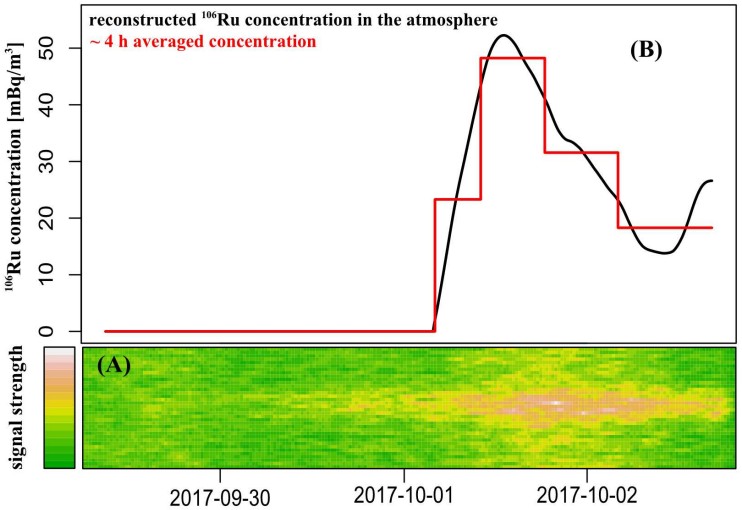

**Figure 2.** The response of the AMARA system to the [106]Ru contamination passing over during the corresponding sampling interval; A) the [106]Ru signal increase in the (615 - 630) keV energy region after subtracting the radon background; B) the example reconstructed real-time [106]Ru concentration and its 4-hour averaged values which corresponds to the CEGAM time resolution .

## 2.2 Real-time sampling and measurement procedure

### 2.2.1 AMARA system

The AMARA system employs a fully continuous measurement regime where the aerosol filter is counted via gamma-ray spectrometry already during sampling using a high-volume (900 m$^3$/h) sampler. In this setup, shown in Fig. 1, a spectrometric module consisting of an HPGe detector is placed directly above the aerosol filter. This straightforward solution benefits from

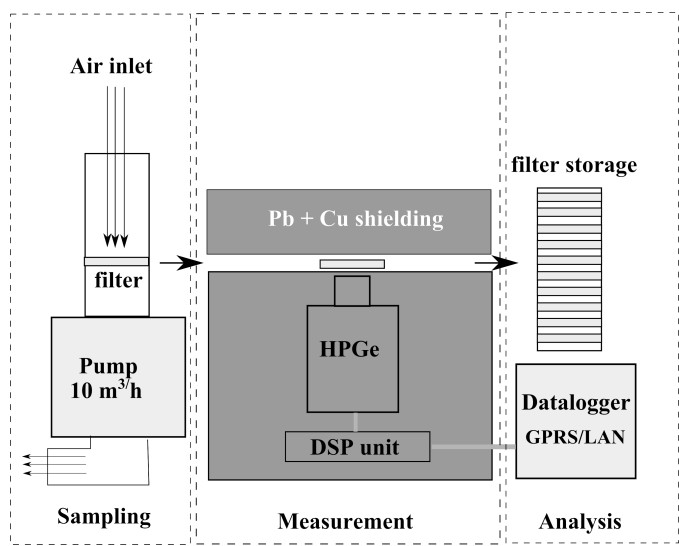

**Figure 3.** CEGAM system schematics; the activity deposition is measured using an HPGe detector above an aerosol filter after sampling in radiation shielding.

its simplicity and from the real-time nature of the measurement. However, the detection limits are higher due to the very high and variable natural background caused mainly by $^{222}$Rn and $^{220}$Rn decay products. Our approach for suppressing the high and widely variable radon background is based on the NASVD algorithm(Minty and Hovgaard, 2002) and consists of extracting the characteristic spectral shapes from a large dataset of background measurements. We adopted this approach already in the

previous version of the AMARA system, which was based on a NaI(Tl) detector. The implementation details are described by Hýža and Rulík (2017) and a demonstration of the signal treatment is displayed in Fig.2.

### 2.2.2   CEGAM system

The CEGAM system is based on semi-continuous sampling where samples are taken at preset intervals and then measured via gamma spectrometry. The device is based on a carousel sampling changer, which moves the aerosol filters between the

sampling position and the measuring position, see the configuration in Fig. 3. This allows the CEGAM's HPGe spectrometer to be placed inside a heavy lead shielding and it is also possible to let the radon progenies decay before the measurement. The natural background level is therefore much lower in comparison with the AMARA system and it yields similar MDAC but at a much lower flow rate (10 m$^3$/h).

### 2.2.3   Measurement procedure and systems comparison

Both the AMARA and CEGAM systems employ an electrically cooled HPGe (ORTEC/CANBERRA) detector in a temperature-/humidity-controlled environment in order to ensure smooth continuous operation even during demanding weather conditions. The signal processing is done by digital multi-channel analyzer (DSPEC/LYNX). The eventual gain shift is automatically

corrected by stabilization algorithm based on the position of background peaks. The efficiency calibrations were done experimentally using aerosol filters spiked with standard activity solutions provided by the Czech metrological institute. For the purpose of calibration and measurement respectively, the correction to the True coincidence summation was taken into account. As the AMARA system operates in a continuous regime, the spectrum acquisition time was set to 5 minutes in order

to make full use of its time resolution. Consequently, the running spectral sums of arbitrary lengths can be constructed. The actual activity values are then computed using the numerical derivative of smoothed cumulative response. On the other hand, the time resolution of the CEGAM system is limited by the carousel changer time steps. Typically, the spectrum acquisition time is set to 24 h and in case of emergency it is shortened to 4 h or less.

The inherent time resolution of the monitoring system is inevitably related to the accuracy of the contamination arrival time.

For the $^{106}$Ru case, the AMARA system estimated its arrival with approx. 1-h accuracy depending on the chosen level of statistical significance and the type of statistical test.

Although the detector efficiency and flow rate are determined relatively accurately, there are other effects negatively influencing the final activity uncertainty. For instance, the radon decay products concentration and therefore the MDAC and the activity uncertainty vary significantly. In case of positive detection, there is also an additional uncertainty contribution due to

the deposition dynamics as the system needs to subtract the contribution from the already deposited contamination. Comparing the real time values with those obtained by laboratory measurements ($^{106}$Ru case or natural $^7$Be) we estimate the uncertainty of (10 – 15) percent for the 4-hour integration time and the activity of several mBq/m$^3$.

Although both systems are intended for a rapid detection of artificial radionuclides in the air, they differ in their typical use. The CEGAM system is an autonomous system with a high filter capacity and it is suitable for remote places with a difficult

access of the operating personnel. The power consumption is also much lower in comparison with the AMARA system due to the employment of a low-volume sampler with an adjustable throughput. During a normal situation, the CEGAM system could be used within a monitoring network as a stand-by device (low flow rate, long sampling intervals) which could quickly switch to an emergency mode (higher flow rate, more frequent sampling). The switching command could be based on some prior information about arriving contamination or on the positive detection in a laboratory or by a more sensitive/ rapid device,

such as AMARA system.

The AMARA system is intended as an upgrade of an already existing monitoring site equipped with a high-volume sampler with operational personnel because the filters are not changed automatically. The advantage of this approach is a better time resolution and therefore a rapid response. Monitoring sites with high volume samplers are usually equipped with a gamma-ray spectrometry laboratory and therefore the filters from AMARA are consequently measured in a dedicated counting room

and potentially investigated further by radiochemical procedures to determine the activities of non gamma-ray emitters. The proximity of laboratory also solves to a certain degree the dilemma between the sensitivity of measurement and sampling duration as the final most sensitive measurement will be performed in laboratory after the sampling using the standard analytic procedure.

Both systems together provide a very good solution for rapid radiation monitoring response to various release scenarios. The

technical parameters are summarized in Tab. 1.

| Parameter | AMARA | CEGAM |
|---|---|---|
| Detector type | HPGe (electrically cooled) | HPGe (electrically cooled) |
| Rel. efficiency | 30 % | 50 % |
| FWHM | 1.9 keV | 1.9 keV |
| Shielding | None | 10 cm Pb |
| Filter size | 57 cm $\times$ 27 cm | 10 cm diameter disc |
| Filter material | FPM 1545 | GF/A glass microfibre |
| Spectrum stabilization | Automatic | Automatic |
| Mode of operation | Continuous | Carousel type sample changer |
| MDAC | $\sim$ mBq/m$^3$ * | $\sim$ mBq/m$^3$ ** |
| Flow rate | 900 m$^3/h$ | 0 to 10 m$^3/h$ |
| Filter cartridge capacity | No cartridge | 300 filters |

\* one-hour integration time and 12 hours of sampling

\*\* per 4-hour sampling/measurement period

**Table 1.** Technical specification of AMARA and CEGAM systems.

## 2.3 Dataset description

The monitoring data comes from 10 standard monitoring sites in the Czech Republic from the time period between 25 September 2017 and 13 October 2017. Once $^{106}$Ru was confirmed by the AMARA system (located in Prague), the filters were changed, and the monitoring interval was shortened at all monitoring sites. The previous, less sensitive version of the AMARA system equipped with NaI(Tl) spectrometer operated in the Hradec Kralove location. Unfortunately, the CEGAM system was not yet operational during the $^{106}$Ru incident , hence, all used data come from AMARA system.

A total of 47 samples were collected, and 24 of them were positive results with reported activity above the MDAC level. Four distinct datasets were derived on the basis of this monitoring campaign:

1. RAW dataset – raw monitoring, as reported by the individual standard monitoring sites. The real-time measurements are not included.

2. WEEKS dataset – derived from the raw dataset by weekly averaging. This dataset corresponds to the standard RMN monitoring regime.

3. FAST dataset – raw data complemented by real-time values from the AMARA and CEGAM (simulated) systems. The integration window was set within the interval of 3 – 13 hours during the concentration peak period.

4. CUT dataset – created by cutting off the time interval between the start of sampling and the arrival of the $^{106}$Ru contamination at the particular monitoring site. As there was no real-time measurement apart from the Prague and Hradec

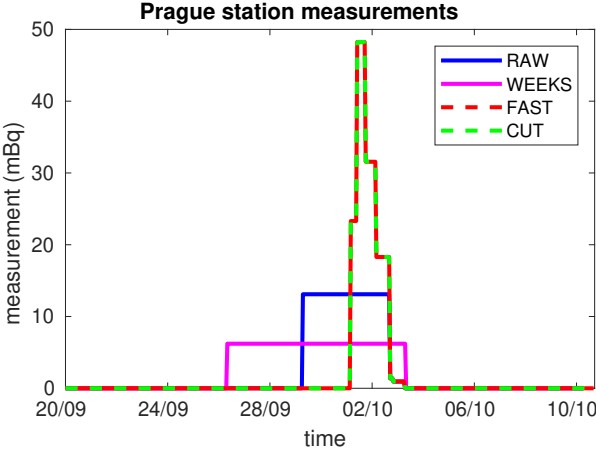

**Figure 4.** The measurements from the Prague station are displayed for each dataset using coloring given in the legend.

Kralove AMARA measurements, the arrival times were estimated on the basis of an overall analysis of the atmospheric transport across the Czech Republic, using the HYSPLIT model.

Note that the artificial WEEKS and CUT datasets are derived from the RAW and FAST datasets, and are rather experimental. All four datasets are attached as a supplement to this article.

For illustration, the measurements from the Prague station (equipped by the AMARA system) are given in Fig. 4 where much better temporal specificity is demonstrated.

## 3   Inverse modeling

The general purpose of inverse atmospheric modeling is to estimate the time profile of an unknown emission, called the source term, in the so-called top-down approach (Nisbet and Weiss, 2010), where ambient measurements are combined with the result

of an atmospheric transport model (ATM). The source term can be estimated using optimization of the differences between the measurements and the corresponding simulated values predicted by an ATM. An even more challenging task is to identify the location of the release. This can be done, e.g., using possible source location selection and comparison, as in the case of the $^{131}$I release in January/February 2017 (Masson et al., 2018), using computed correlation or cost function maps as in the case of radioxenon after the third North Korea nuclear test (De Meutter et al., 2018), or using a Bayesian approach as in the case

of the $^{131}$I release in the fall of 2011 (Tichý et al., 2017) or in the case of the $^{75}$Se leakage in 2019 (De Meutter and Hoffman, 2020).

In this paper, we follow the general concept of a linear model of the atmospheric dispersion using an SRS matrix (e.g., Seibert (2001); Seibert and Frank (2004)). Here, an atmospheric transport model is used to calculate the linear relation between the potential source and the measured concentrations. Aggregating all possible time steps of the release in a source term vector

$\mathbf{x} \in \mathbf{R}^n$ and measurements from all sites and times in the vector $\mathbf{y} \in \mathbf{R}^p$, we can define the model

$$\mathbf{y} = \mathbf{M}\mathbf{x} + \mathbf{e}, \tag{1}$$

where $\mathbf{M} \in \mathbf{R}^{p \times n}$ is the SRS matrix and $\mathbf{e} \in \mathbf{R}^p$ is an observation error, where the model errors and the measurements errors are aggregated. This concept has been largely used previously to recover the source term within larger-scale scenarios such as nuclear power plant accidents (Stohl et al., 2012; Evangeliou et al., 2017), estimates of the emission of greenhouse gases (Stohl et al., 2009), or volcanic emission (Kristiansen et al., 2010).

The estimation of the source term vector $\mathbf{x}$ from Eq. (1) is non-trivial, since the SRS matrix $\mathbf{M}$ is typically ill-conditioned and some regularization is needed. One possible approach is to minimize a suitable cost function (Eckhardt et al., 2008; Evangeliou et al., 2017) such as

$$J = (\mathbf{y} - \mathbf{M}\mathbf{x})^T \mathbf{R} (\mathbf{y} - \mathbf{M}\mathbf{x}) + \mathbf{x}^T \mathbf{B}\mathbf{x} + \epsilon \mathbf{x}^T \mathbf{D}^T \mathbf{D}\mathbf{x}, \tag{2}$$

The first term stands for the deviation of the model from the measurement, including the error in the meteorological data; the second term penalizes high values of the source term using diagonal matrix $\mathbf{B}$; and the third term favors the smoothness of the estimated source term using tridiagonal matrix $\mathbf{D}$ (numerically representing the second derivative) and weighting coefficient $\epsilon$. The key issue of the minimization is then to select matrices $\mathbf{R}$, $\mathbf{B}$, and $\epsilon$.

The minimization of Eq. (2) can be interpreted using a probabilistic model and the proper Bayesian inference can be used to estimate the source term $\mathbf{x}$. Consider the logarithm of the likelihood function

$$\ln p(\mathbf{y}|\mathbf{x}, \mathbf{R}) = \ln \mathcal{N}(\mathbf{M}\mathbf{x}, \mathbf{R}^{-1}) \propto (\mathbf{y} - \mathbf{M}\mathbf{x})^T \mathbf{R} (\mathbf{y} - \mathbf{M}\mathbf{x}), \tag{3}$$

where symbol $\propto$ denotes equality up to the normalizing constant, then $\ln p(\mathbf{y}|\mathbf{x}, \mathbf{R})$ is the probabilistic equivalent to the first term of $J$. Equivalents for the second term and for the third term can be found in a similar way. However, one benefit of the Bayesian inference is that the elements of $\mathbf{R}$, $\mathbf{B}$, and $\epsilon$ do not need to be fixed in advance but can also be estimated and optimized within the method. The second benefit is the model selection property of the Bayesian inference (Bernardo and Smith, 2009). This approach can be used to select the most likely setting of the dispersion model or the most likely matrix $\mathbf{M}$ when it is computed for multiple locations (Tichý et al., 2017).

In the following sections, we review the Bayesian inversion method based on similar probabilistic formulation as in Eq. (3) called the least squares with adaptive prior covariance (LS-APC) (Tichý et al., 2016). We then discuss an extension of the method using a covariance model of the measurements.

### 3.1 Probabilistic LS-APC model

The probabilistic inversion model of Tichý et al. (2016) called LS-APC (least squares with adaptive prior covariance) is briefly reviewed and its extension is discussed. In (Tichý et al., 2016), the covariance structure has been simplified as $\mathbf{R} = \omega \mathbf{I}$, where $\mathbf{I}$ is the identity matrix. This simplification may be misleading. We therefore consider the likelihood in Eq. (3) with covariance $\mathbf{R}$ scaled by the scalar parameter $\omega$ being considered unknown. In the variational Bayes inference, all unknown parameters

need to be accompanied by their prior distribution. We select gamma distribution for $\omega$ due to its conjugacy with the Gaussian likelihood (Tipping and Bishop, 1999) obtaining the data model in the form:

$$p(\mathbf{y}|\mathbf{x},\omega) = \mathcal{N}\left(\mathbf{Mx}, \omega^{-1}\mathbf{R}^{-1}\right), \tag{4}$$

$$p(\omega) = \mathcal{G}\left(\vartheta_0, \rho_0\right), \tag{5}$$

where $\vartheta_0, \rho_0$ are selected constants needed for numerical stability; however, they are selected very low, e.g. $10^{-10}$, providing a non-informative prior. The construction of the precision matrix $\mathbf{R}$ (inverse covariance) will be discussed in the next section.

    The prior model of $\mathbf{x}$ is a probabilistic relaxation of the second and third terms in Eq. (2). The prior is chosen to be Gaussian truncated to positive support (notation $t\mathcal{N}(\mu, \sigma, [a, b])$, see (Tichý et al., 2016) for details) with a covariance matrix in the specific form of the Cholesky decomposition

$$p(\mathbf{x}|\mathbf{\Upsilon}, \mathbf{L}) = t\mathcal{N}\left(\mathbf{0}, \left(\mathbf{L}\mathbf{\Upsilon}\mathbf{L}^T\right)^{-1}, [0, +\infty]\right), \tag{6}$$

where $\mathbf{\Upsilon}$ is a diagonal matrix with diagonal entries $\upsilon_j$ and $\mathbf{L}$ is a lower bidiagonal matrix with ones on the diagonal and sub-diagonal entries $l_j$. The prior models for the unknowns $\upsilon_1, \ldots, \upsilon_n$ and $l_1, \ldots, l_{n-1}$ are selected as

$$p(\upsilon_j) = \mathcal{G}\left(\alpha_0, \beta_0\right), \tag{7}$$

$$p(l_j|\psi_j) = \mathcal{N}\left(-1, \psi_j^{-1}\right), \tag{8}$$

$$p(\psi_j) = \mathcal{G}\left(\zeta_0, \eta_0\right), \tag{9}$$

where parameters $\upsilon_j$ model the sparsity of the source term $\mathbf{x}$ and parameters $l_j$ model the smoothness using prior selection of the mean value as $-1$. The prior constants $\alpha_0, \beta_0$ are selected similarly to Eq. (5) as $10^{-10}$, while the prior constants $\zeta_0, \eta_0$ are selected as $10^{-2}$ to favor a smooth solution, see the discussion in Tichý et al. (2016) for more details. We also note that the algorithm is shown to be robust with respect to the choice of starting and tuning parameters, see discussion in Tichý et al.
(2020) for more details.

    The key parameter in the inversion method, which has not yet been discussed, is the error covariance matrix $\mathbf{R}$ in Eq. (4). The definition of this matrix will be given and will be discussed in the next section.

## 3.2   Measurement error covariance

There are various approaches in the literature for selecting the shape of the covariance matrix $\mathbf{R}$. A straightforward assumption
is the diagonal model with the same (Tichý et al., 2016; Liu et al., 2017) entries where this scalar value can be estimated. When considering different entries on the diagonal of $\mathbf{R}$, they may be selected on the basis of physical information, when available, rather than estimating them, because numerical issues arise during convergence (Berchet et al., 2013). A common assumption is to compose the diagonal entries from three source of errors: (i) the absolute error of the measurement, (ii) the relative error of the measurement, and (iii) the application dependent error, such as the model-observation mismatch (Brunner et al., 2012;
Song et al., 2015) or the error based differences between observations and simulations (Henne et al., 2016).

Similarly to (Stohl et al., 2012; Evangeliou et al., 2017), we adopt the first two error terms in our covariance structure while introducing the third term based on the length of the measurement. In sum, the $\mathbf{R}$ is

$$\mathbf{R} = \text{diag} \sqrt{\boldsymbol{\sigma}_{\text{abs}}^2 + (\boldsymbol{\sigma}_{\text{rel}} \circ \mathbf{y})^2 + \frac{1}{\boldsymbol{\sigma}_{\text{length}}^2}}, \tag{10}$$

where $\boldsymbol{\sigma}_{\text{abs}}^2$ is the absolute measurement error which is selected between 0.2 and 1.4 mBq based on the maximum a posteriori estimate, $\boldsymbol{\sigma}_{\text{rel}}$ is the uncertainty level of measurements, which is between 5.5 and 30% for our dataset, and $1/\boldsymbol{\sigma}_{\text{length}}^2$ is the term considering the length of the measurement as $\boldsymbol{\sigma}_{\text{length}} = \frac{\text{measurement hours}}{6}$ where the selection of 6 hours window is motivated by the GFS meteorological data resolution. Here, a shorter measurement time implies higher uncertainty and a longer measurement time implies lower uncertainty.

## 3.3 Variational Bayes inference and source location

Within the variational Bayes (VB) framework (Šmídl and Quinn, 2006), the posterior distributions are found in the same functional form as their priors. The moments of the posteriors are determined using an iterative algorithm with details in Tichý et al. (2016). Here, the reference Matlab implementation can be downloaded as a supplement. The method will be denoted here as the LS-APC-VB method.

Moreover, we consider the scenario where we have a finite set of SRS matrices $\{\mathbf{M}_1, \mathbf{M}_2, \ldots, \mathbf{M}_r\}$, representing different considered locations of the release here. For each SRS matrix from the set, we can evaluate the posterior probability $p(\mathbf{M} = \mathbf{M}_k | \mathbf{y})$ as

$$p(\mathbf{M} = \mathbf{M}_k | \mathbf{y}) \propto p(\mathbf{M} = \mathbf{M}_k) \exp(\mathcal{L}_{\mathbf{M}_k}), \ k = 1, \ldots, r, \tag{11}$$

where $p(\mathbf{M} = \mathbf{M}_k)$ is the prior probability of $\mathbf{M}_k$ which can be omitted here since each location has the same prior probability and $\mathcal{L}_{\mathbf{M}_k}$ is a variational lower bound on $p(\mathbf{y}|M_k)$ (Bishop, 2006). Finally, the term $\mathcal{L}_{\mathbf{M}_k}$ can be computed as (Tichý et al., 2017)

$$\mathcal{L}_{\mathbf{M}_k} = \text{E}\left[\ln p(\mathbf{y}, \mathbf{x}, \Upsilon, \mathbf{L}, \boldsymbol{\psi}, \omega, \mathbf{M}_k)\right] - \text{E}\left[\ln \tilde{p}(\omega)\right] - \text{E}\left[\ln \tilde{p}(\mathbf{x})\right] - \text{E}\left[\ln \tilde{p}(\Upsilon)\right] - \text{E}\left[\ln \tilde{p}(\mathbf{L})\right] - \text{E}\left[\ln \tilde{p}(\boldsymbol{\psi})\right], \tag{12}$$

where $\text{E}[.]$ denotes the expected value with respect to the distribution of the variable in its argument and $\tilde{p}()$ are approximate posterior probability distributions. These terms are given in the supplementary material of (Tichý et al., 2017).

Note that to display and to compare the computed probabilities for each computational domain in following sections, we need to normalize results due to the proportional equality in Eq. (11). We use normalization using maximum of each domain so that the maximum of each normalized domain is equal to 1.

## 4 Experiments and discussion

The aims of our experiments are to estimate the location of the $^{106}$Ru source, to estimate the source term, and to compare results obtained using four datasets from the Czech monitoring network introduced in Section 2, and with results obtained

using the dataset reported by the International Atomic Energy Agency (IAEA) (IAEA, 2017). For this purpose, we use the HYSPLIT atmospheric transport model (Stein et al., 2015; Draxler and Hess, 1997), coupled with the NCEP/NOAA global forecast system (GFS) meteorological data.

To validate our results, we also use the FLEXPART model (Pisso et al., 2019) coupled with meteorological analyses from the European Centre for Medium-Range Weather Forecasts (ECMWF) to study the release based on location selected using HYSPLIT model simulations.

## 4.1 Atmospheric transport modeling

### 4.1.1 HYSPLIT model configuration

We use the HYSPLIT model in backward mode to compute all the required SRS matrices for a domain. The spatial domain is selected to cover the region spanning from $5°$ E to $115°$ E in longitude and from $25°$ N to $65°$ N in latitude, covering Central and Eastern Europe and the western half of the Russian Federation. Note that the displayed domain in the following figures is cropped in order to focus on the important area only. Spatially, the domain was discretized with resolution $0.5° \times 0.5°$. Vertically, there is no discretization of the domain, and sensitivities are calculated for a layer 0–300 m above the ground, which allows for both ground releases and somewhat elevated releases, e.g. through a stack. The temporal resolution is selected as 6 hours, starting from 20 September and ending on 10 October 2017. Runs were forced with GFS meteorological fields with horizontal resolution of $0.5° \times 0.5°$, 26 vertical layers, and 6 hours temporal resolution.

The SRS matrices for the domain are computed from HYSPLIT backward runs for each domain grid cell. The backward run configuration is selected since the number of domain grid cells (17600) is much higher than the number of measuring sites (tens, depending on the dataset, hundreds in the case of the IAEA dataset). Each backward run starts at the point location of each measuring site and release particles during the period corresponding to the measurement time of the sample. For each run, 1 million particles were simulated. Each of the backward runs corresponding to one measurement provides an SRS field of a particular measurement to all spatial-temporal sources in the selected domain. We assume that the release occurred from a point source, and that we can therefore calculate SRS matrices for the whole domain at once. We end up with 17600 SRS matrices for each dataset, all of which are source location candidates.

### 4.1.2 FLEXPART model configuration

FLEXPART version 10.4 (Pisso et al., 2019) releases computational particles that are tracked in time following 3-hourly operational meteorological analyses from the European Centre for Medium-Range Weather Forecasts (ECMWF) with 137 vertical layers and a horizontal resolution of $1° \times 1°$. The model accounts for dry and wet deposition (Grythe et al., 2017), turbulence (Cassiani et al., 2015), unresolved mesoscale motions (Stohl et al., 2005) and convection (Forster et al., 2007). SRSs were calculated for 30 days backward in time, at temporal intervals that matched measurements at each receptor site. [106]Ru is tracked assuming gravitational settling for spherical particles with an aerosol mean diameter of 0.6 $\mu$m and a normalised standard deviation of 3.3 and a particle density of 2500 kg m$^3$ (Masson et al., 2019).

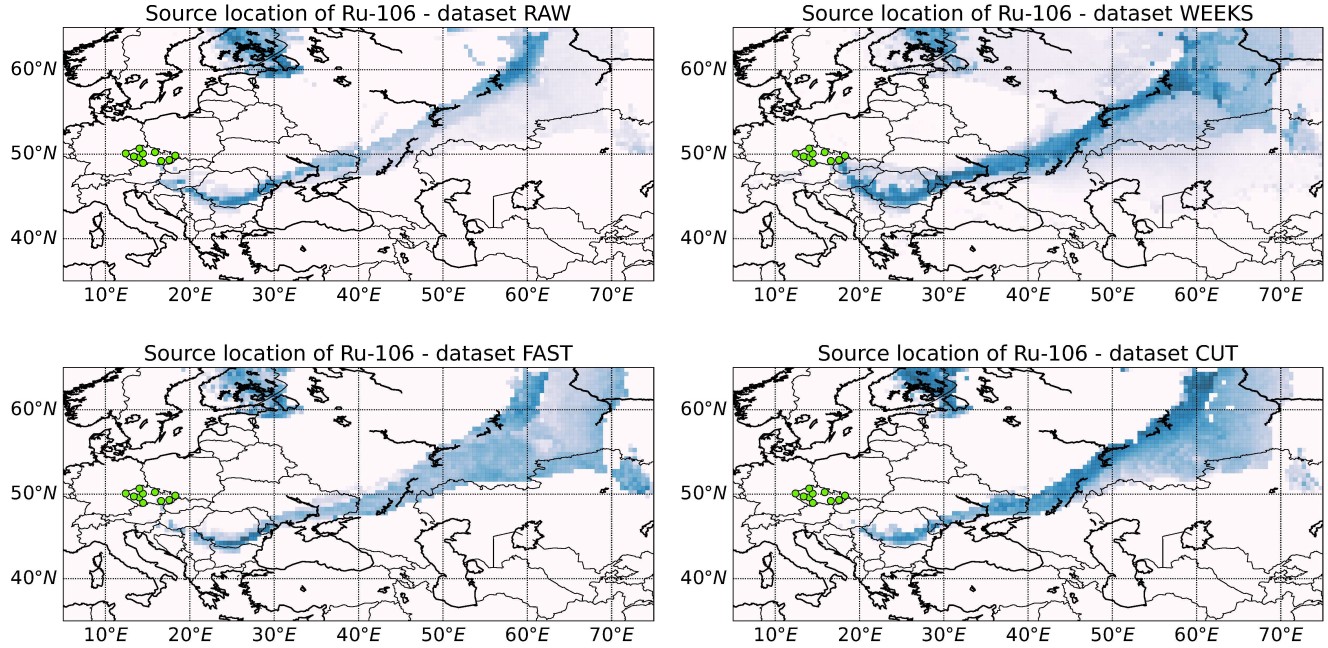

**Figure 5.** Source location of the [106]Ru release via the marginal log-likelihood, where the observed data are explained by a release from a grid cell using the LS-APC-VB method HYSPLIT atmospheric transport model coupled with GFS 0.5° meteorological data. The dataset that has been used is indicated in the titles of each map. The measuring sites are displayed using green dots.

## 4.2 Results for the Czech monitoring data

For each dataset and each SRS matrix, we apply the LS-APC-VB method to compute the probability of each spatial grid cell according to Eq. (11). Note that no prior information on source location, $p(\mathbf{M} = \mathbf{M}_k)$, in Eq. (11) is used. This corresponds to the assumption that all locations are equally possible. The resulting maps with source location probabilities for the RAW (top left), WEEKS (top right), FAST (bottom left), and CUT (bottom right) datasets are displayed in Fig. 5. Here, a darker color means a more probable location of the release while the scale is relative and dimensionless due to the proportional equality in Eq. (11).

In all four cases, an estimated probability region of source locations forms the strip spanning from southern Romania to approximately the Ob river in the Russian Federation. Notably, these regions are computed on the basis of data from the Czech monitoring stations only. Limited ability of the method to determine one specific location was therefore expected. During the period in question, the wind mostly blew towards the west, which is in agreement with the probable source region located to the east of the Czech republic. The RAW dataset tends to prefer the northern part of the estimated source location strip, leaving the south part less probable. Similar behavior is observed in the case of the WEEKS dataset where, in addition, low probability was also observed in wide areas in the south and north of the strip. This is probably caused by the lower temporal

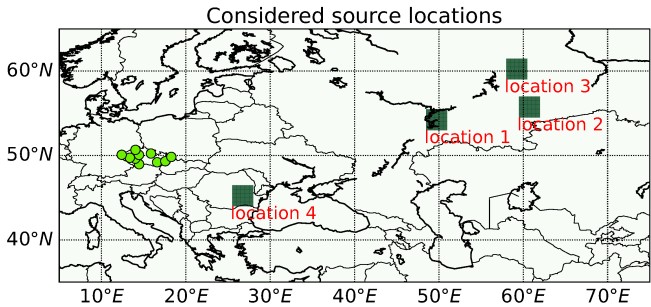

**Figure 6.** The four considered locations are displayed using green squares and labels. The measuring sites are displayed using green dots.

| study | probable source location | total release | temporal character (year 2017) |
|---|---|---|---|
| (Kovalets and Romanenko, 2017) | Urals, southern Russia | 1 TBq to 1 PBq | – |
| (Sørensen, 2018) | Dimitrovgrad or Mayak | $< 1.1$ PBq | 26 September, between 5:00 and 13:00 (Mayak) |
| (De Meutter et al., 2019) | Mayak | $< 1$ PBq | – |
| (Maffezzoli et al., 2019) | Mayak | – | – |
| (Shershakov et al., 2019) | Mayak | $\sim 500$ TBq | 25 and 26 September |
| (Saunier et al., 2019) | Mayak | $250 \pm 13$ TBq | 26 September (small activity also on 23 and 24 September) |
| (Le Brazidec et al., 2020) | Mayak | between 100 and 200 TBq | 26 September |
| (Western et al., 2020) | Mayak | $441 \pm 13$ TBq | 24 September, between 12:00 and 18:00 |
| Source term based on Czech FAST dataset | Mayak | $237 \pm 107$ TBq | between 6:00 AM on 25 September and 6:00 AM on 26 September |

**Table 2.** This table summarizes and compares previous studies on the [106]Ru release in 2017, focusing on the total release, the source location, and the temporal character. The last row contains results based on the Czech FAST dataset.

resolution of the measurements, implying a wider possibility of radionuclide transport. The results obtained using the FAST and CUT datasets are more homogeneous, covering the whole strip. However, the CUT dataset provides locations with very low probability inside the strip. These are probably artifacts caused by the artificial adjustment of the data. Note that better source location is possible with better spatial distribution of the measuring sites. This is, indeed, available and will be discussed in Sec. 4.3 on the IAEA dataset.

Based on Fig. 5 and a review of the situation in the literature, see Tab. 2, we consider four source locations. Two of them are Russian nuclear facilities capable of producing a significant amount of [106]Ru (Saunier et al., 2019; Masson et al., 2019;

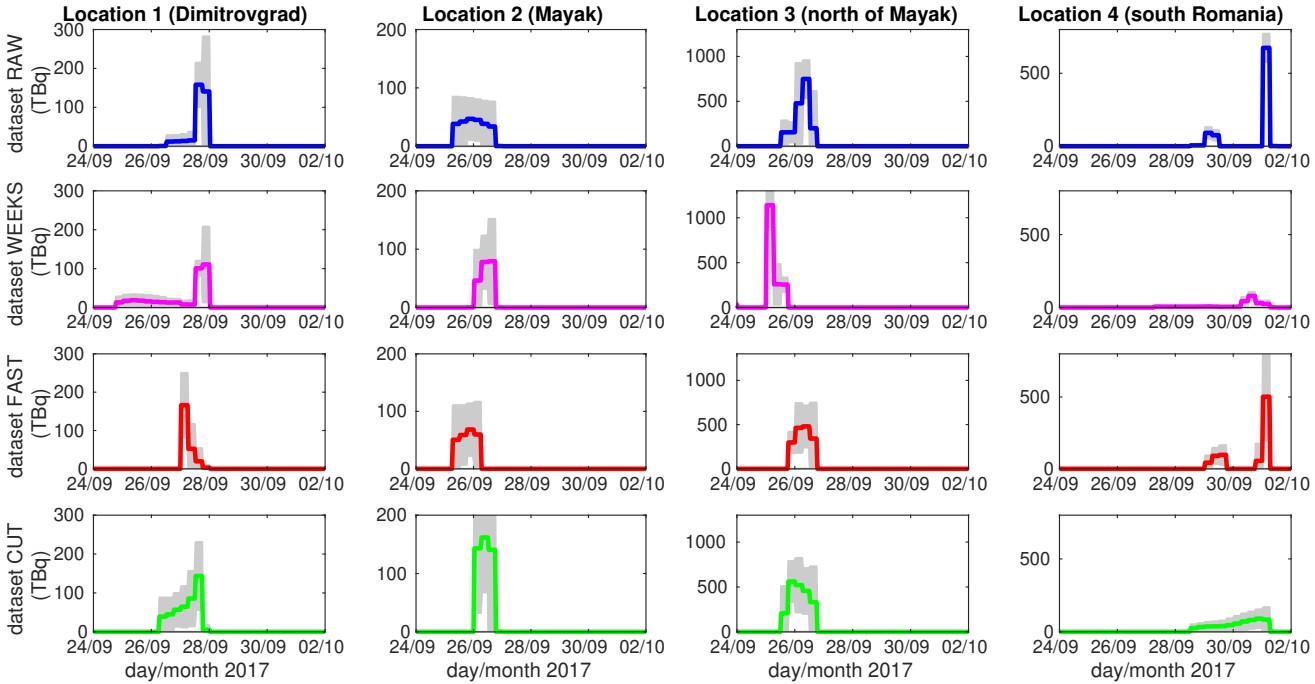

**Figure 7.** Estimated source terms from the locations considered in Fig. 6 (indicated in the titles of each column) for the RAW (blue lines), WEEKS (magenta lines), FAST (red lines), and CUT (green lines) datasets. The estimated source terms are accompanied by the 95% uncertainty regions (gray filled regions). Note that the vertical axis has a different scales for each location.

Sørensen, 2018): the Research Institute of the Atomic Reactor (RIAR) in Dimitrovgrad (location 1) and the Mayak Production Association, a spent fuel reprocessing facility in Ozersk (location 2), see Fig. 6. Location 3 is selected as a location with

high probability in all four datasets, and is situated to the east of Perm, to the north of the Mayak location. Location 4 is situated in southern Romania, and is also a candidate according to all datasets. We are aware that, according to further analyses (Le Brazidec et al., 2020; Saunier et al., 2019; Shershakov et al., 2019; De Meutter et al., 2019; Western et al., 2020), all locations except Mayak, location 2, could be rejected. However, we have considered them here, since they are candidate locations based on just Czech monitoring data. Dimitrovgrad, location 1, was later rejected due to inconsistency with the

concentration measurements to the south and east of Dimitrovgrad (Saunier et al., 2019; Maffezzoli et al., 2019). Location 3 is hypothetical, with no known nuclear facility around the location capable of producing a substantial amount of [106]Ru that would explain the concentration measurements thousands kilometers away from this location. A release at location 4 in southern Romania would contradict ground-based observations tothe east of the location was thus also rejected (see Masson et al. (2019)). Nevertheless, we will discuss all four possible source terms in these locations in this Section, in order to demonstrate

the effects of the fast measuring systems.

| estimated total ST (TBq) | RAW | WEEKS | FAST | CUT |
|---|---|---|---|---|
| location 1 (Dimitrovgrad) | 352 | 363 | 241 | 439 |
| location 2 (Mayak) | 245 | 203 | 237 | 445 |
| location 3 (north of Mayak) | 1737 | 1755 | 1583 | 2075 |
| location 4 (south Romania) | 853 | 248 | 787 | 603 |

**Table 3.** Estimated total source terms in TBq for a specific dataset (columns) and for a specific location (rows).

| estimated length (hours) | RAW | WEEKS | FAST | CUT |
|---|---|---|---|---|
| location 1 (Dimitrovgrad) | 36 | 78 | 24 | 42 |
| location 2 (Mayak) | 36 | 18 | 24 | 18 |
| location 3 (north of Mayak) | 30 | 42 | 24 | 30 |
| location 4 (south Romania) | 30 | 96 | 30 | 66 |

**Table 4.** Estimated length of non-zero activity (higher than 1 TBq in a period of 6 hours) of source terms in hours for a specific dataset (columns) and for a specific location (rows).

The estimated source terms are displayed in Fig. 7 for all the considered datasets and locations, see the titles and labels. Note that in Fig. 7 we have cropped zero activities at the beginning and at the end of the source terms to maintain better visibility. All source terms are associated with the 95% (two sigmas) highest posterior density region, using gray-filled regions. The total estimated activities are further summarized in Tab. 3. Note that only the Dimitrovgrad and Mayak locations are in agreement with the previously reported total activities of approximately 100 - 500 TBq (Shershakov et al., 2019; Saunier et al., 2019; Le Brazidec et al., 2020; Western et al., 2020). Estimates from all datasets for these locations fit this interval.

As regards the temporal specification of the release, the estimated lengths of the release are displayed in Tab. 4. The release probably occurred at Mayak between 25 September and 26 September, see literature review in Tab. 2. Shershakov et al. (2019) estimated the two-days interval (both 25 September and 26 September) while further analyses by Saunier et al. (2019) and by Le Brazidec et al. (2020) indicate a higher probability of the release on 26 September, with a possible minor release on 23 September and 24 September (Saunier et al., 2019). This is consistent with our findings, where 26 September was estimated using the WEEKS and CUT datasets; most of both days, 25 and 26 September, were estimated using the RAW dataset; and the time period between 6:00 AMon 25 September and 6:00 AM on 26 September was estimated by the FAST dataset. Further validation with the IAEA dataset, Sec. 4.3, shows that the estimates from the WEEKS and FAST datasets are in better agreement with the IAEA reported concentration measurements than the estimates from the RAW and CUT datasets. Considering that the bulk of the release was probably within one day, we conclude that the FAST dataset provides the most consistent results, estimating a one-day (24 hours) release for locations 1, 2, and 3 and 30 hours for location 4. The RAW dataset estimated that the release lasted between 30 and 36 hours. Wider ranges were obtained in the case of the WEEKS dataset (between 18 and 96 hours) and the CUT dataset (between 18 and 66 hours). This wide ranges of the release from different locations are probably

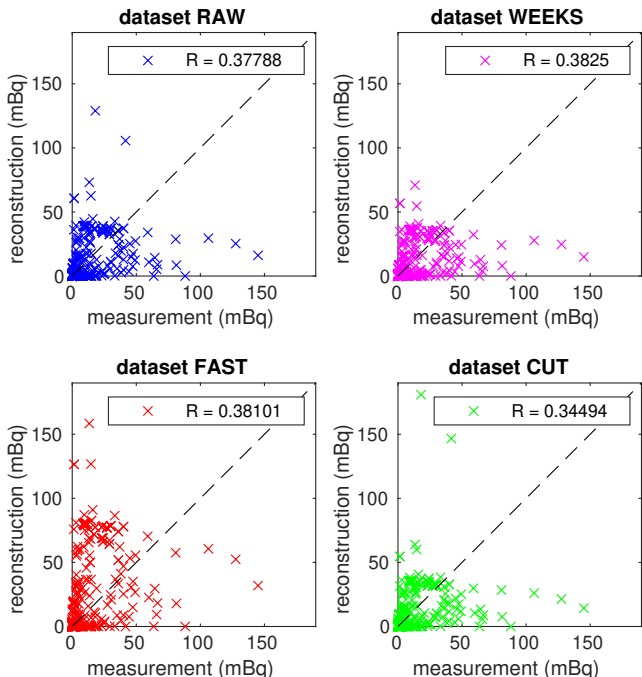

**Figure 8.** Scatter plots between the IAEA measurements and reconstructions using the RAW, WEEKS, FAST, and CUT datasets (specified in titles) for location 2, Mayak. Computed correlation coefficients are given in the legends.

caused by the natural assumption of the LS-APC model that the shorter release is more probable than a longer one using selection a zero prior mean value of the source term in Eq. (6). These findings support the hypothesis that the fast measuring
systems have better time-specificity than the standard measurement procedure.

### 4.3 Validation and comparison with the IAEA dataset

The same atmospheric transport modeling procedure as in Sec. 4.1 is applied here to the dataset of the [106]Ru measurements available from the IAEA report (IAEA, 2017). This consists of 451 relevant measurements, mostly from Northern, Eastern and Central Europe and the Russian Federation, see Fig. 10 for the exact locations of the measuring sites. This dataset will serve as
a validation set (Czech monitoring data has been removed).

First, scatter plots between the measured data reported by the IAEA and s reconstruction using estimated source terms from the four studied Czech datasets studied here are displayed in Fig. 8 for location 2, Mayak. Here, the same colors as in Fig. 7 for each dataset are used. The scatter plots are accompanied by the computed correlation coefficient (R value) given in the legend of each plot. We observed that the highest correlations coefficients are for the WEEKS (0.383) and FAST (0.381)
datasets. The RAW dataset has a lower correlation coefficient (0.378) and the CUT dataset has a significantly lower correlation coefficient (0.345). This demonstrates that the fast measuring systems provide comparable or even better results than the

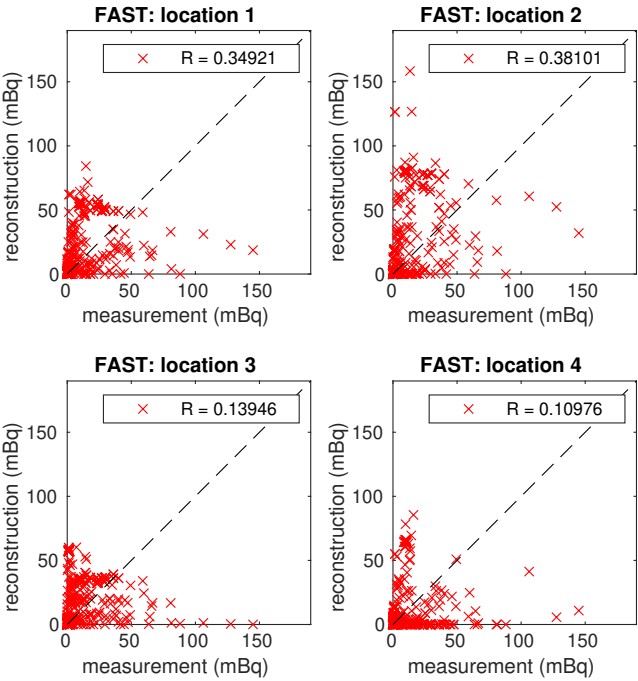

**Figure 9.** Scatter plots between the IAEA measurements and reconstructions using the FAST dataset for all four considered locations (specified in the titles). Computed correlation coefficients are given in the legends.

standard measurement procedure. The artificially constructed CUT dataset has a significantly lower agreement with the IAEA dataset, which may indicate e.g. inaccuracy in cutting the time intervals of the measurements in this dataset.

Second, the scatter plots between the measured data reported by the IAEA and the reconstruction using the FAST dataset for all four considered locations are displayed in Fig. 9, accompanied by the computed correlation coefficients. Here, the reconstruction for location 2 (Mayak) is in better agreement with the IAEA data than any other considered location. Note that similar results are also obtained also for all other datasets, indicating that the Mayak location is the most consistent with the IAEA dataset. This confirms the findings of previous studies (Saunier et al., 2019; Maffezzoli et al., 2019; De Meutter et al., 2018; Le Brazidec et al., 2020), which suggest the Mayak location as the most probable.

Third, similarly as for the Czech monitoring data, the source location methodology from Section 3.3 is also applied to the IAEA dataset. The results are displayed in Fig. 10. Again, a darker color denotes a more likely location of the release, while the scale is relative and dimensionless due to the proportional equality in Eq. (11). In direct comparison with the source locations using the smaller datasets studied in Fig. 5, the patterns are very similar. Indeed, the source location using the IAEA dataset rejected locations that cannot be rejected on the basis of the Czech data alone, due to the lack of data, see e.g. the locations in Romania, Ukraine, and Finland. However, the estimates using all datasets in the southern Urals are consistent with the IAEA dataset results, and also with e.g. the results of Saunier et al. (2019). For a numeric comparison of the source

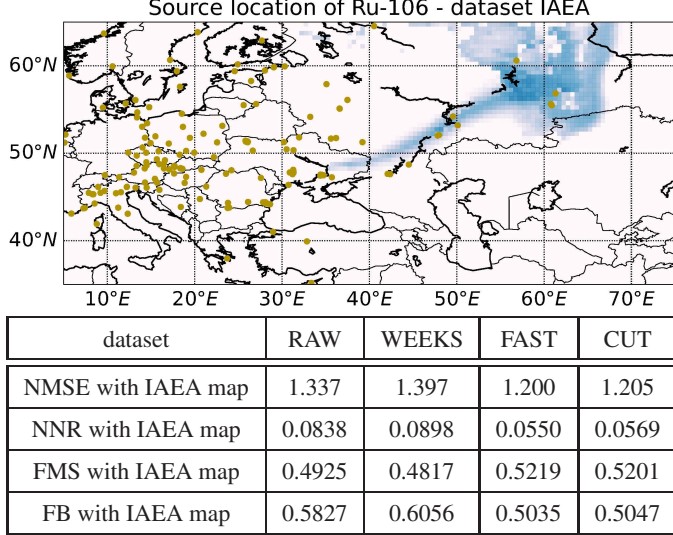

| dataset | RAW | WEEKS | FAST | CUT |
|---|---|---|---|---|
| NMSE with IAEA map | 1.337 | 1.397 | 1.200 | 1.205 |
| NNR with IAEA map | 0.0838 | 0.0898 | 0.0550 | 0.0569 |
| FMS with IAEA map | 0.4925 | 0.4817 | 0.5219 | 0.5201 |
| FB with IAEA map | 0.5827 | 0.6056 | 0.5035 | 0.5047 |

**Figure 10.** Top: source location of the release of $^{106}$Ru via marginal log-likelihood, using the IAEA dataset. Bottom: the computed normalized mean square error (NMSE), the normalized mean square error of the distribution of the normalized ratios (NNR), the figure of merit in space (FMS), and the fractional bias (FB) between the source location results obtained using the IAEA dataset and the RAW, WEEKS, FAST, and CUT datasets.

location maps using the Czech datasets and the map using the IAEA dataset, we compute four statistical coefficients used for evaluations of atmospheric modeling results. Concretely, we compute the normalized mean square error (NMSE) which may be, however, biased (Poli and Cirillo, 1993). Therefore, we also compute the normalized mean square error of the distribution of the normalized ratios (NNR) suggested by Poli and Cirillo (1993) accompanied also by the figure of merit in space (FMS) (Abida and Bocquet, 2009) and the fractional bias (FB) (Chang and Hanna, 2004). Note that coefficients closer to zeros are better in all cases except the FMS where higher is better. These statistical coefficients are defined as

$$\text{NMSE} = \frac{\frac{1}{q}\sum_{j=1}^{q}\left(\mathbf{p}_{\text{IAEA},j} - \mathbf{p}_{\text{set},j}\right)^2}{\left(\frac{1}{q}\sum_{j=1}^{q}\mathbf{p}_{\text{IAEA},j}\right)\left(\frac{1}{q}\sum_{j=1}^{q}\mathbf{p}_{\text{set},j}\right)}, \tag{13}$$

$$\text{NNR} = \frac{\sum_{j=1}^{q}(1-\exp(-|\ln\frac{\mathbf{p}_{\text{IAEA},j}}{\mathbf{p}_{\text{set},j}}|))^2}{\sum_{j=1}^{q}\exp(-|\ln\frac{\mathbf{p}_{\text{IAEA},j}}{\mathbf{p}_{\text{set},j}}|)}, \tag{14}$$

$$\text{FMS} = \frac{\sum_{j=1}^{q}\min(\mathbf{p}_{\text{IAEA},j}, \mathbf{p}_{\text{set},j})}{\sum_{j=1}^{q}\max(\mathbf{p}_{\text{IAEA},j}, \mathbf{p}_{\text{set},j})}, \tag{15}$$

$$\text{FB} = 2\frac{\frac{1}{q}\sum_{j=1}^{q}\mathbf{p}_{\text{IAEA},j} - \frac{1}{q}\sum_{j=1}^{q}\mathbf{p}_{\text{set},j}}{\frac{1}{q}\sum_{j=1}^{q}\mathbf{p}_{\text{IAEA},j} + \frac{1}{q}\sum_{j=1}^{q}\mathbf{p}_{\text{set},j}}, \tag{16}$$

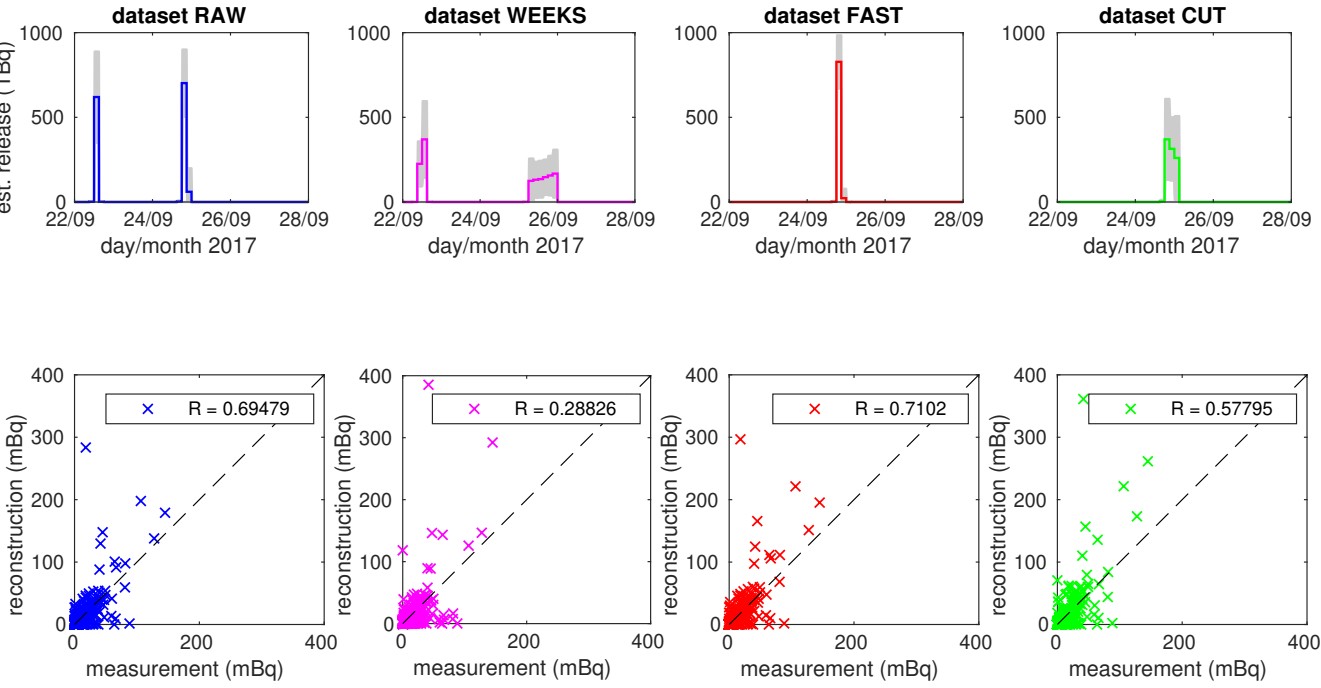

**Figure 11.** Estimated source terms for location 2, Mayak, using SRS matrices computed using FLEXPART atmospheric transport model (top row) associated with scatter plots between the IAEA reported measurements and reconstruction using specified dataset (bottom row). The coloring of panels is the same as in previous Fig. 7 and Fig. 8.

where $q$ is the number of map tiles, $\mathbf{p}_{\mathrm{IAEA}}$ is the vector with the probabilities of the source location computed using the IAEA dataset, and $\mathbf{p}_{\mathrm{set}}$ is the vector with the probabilities of the source location computed using the selected Czech dataset. The results are summarized in Fig. 10, below the probability map.

We conclude that in all cases, results obtained using the FAST dataset are better than those obtained using other datasets. The CUT dataset performs slightly worse than the FAST dataset while results obtained using RAW and WEEKS datasets are significantly worse than obtained using FAST and CUT datasets. This demonstrates that the use of fast measurement systems could better reflect the variability of the release even when it is located far from the release site, and could better match the results of the IAEA dataset, which has a far better spatial distribution of the measurement stations.

## 4.4 Results using FLEXPART model

In this section, we aim to demonstrate that better time-resolution of measurement is beneficial independently on the used atmospheric transport model and the used time-resolution. Concretely, we use FLEXPART model (Pisso et al., 2019) in backward mode with finer, 3 hours, output temporal resolution as described in Sec. 4.1.

We present results for considered location 2, Mayak, in Fig. 11. There are source terms estimated using the LS-APC algorithm in the top row and scatter plot between measured data reported by the IAEA and reconstruction using each dataset in the bottom row. Note that the coloring of panels is the same as in previous Fig. 7 and Fig. 8. The totals of source terms are 1388 TBq, 1459 TBq, 852 TBq, and 948 TBq for datasets RAW, WEEKS, FAST, and CUT respectively. The lengths of releases are 18 hours, 24 hours, 9 hours, and 12 hours for datasets RAW, WEEKS, FAST, and CUT respectively.

Following differences are observed in comparison with results based on HYSPLIT runs. First, we observe significant releases between the 22th and the 23th September in the case of RAW and WEEKS datasets. These releases are not observed for FAST and CUT datasets. However, note also that the response on this initial release in, e.g., IAEA dataset is relatively low, see comparison of R values in Fig. 11. Second, the release periods are estimated rather in the beginning of the 25th September than in the end as in the case of HYPSLIT runs, however, this difference is negligible considering the temporal-spatial domain. Third, totals of releases are in all cases significantly larger than in the case of HYSPLIT runs. The reason for this disproportion may be in different parametrization of the atmospheric model. Considering the scatter plots in Fig. 11, bottom, we assume the estimated releases slightly overestimated while they are on the upper limit of estimates in literature, Tab. 2.

From this perspective, the better temporal resolution of the output temporal grid seems to better reflect better temporal resolution of the measurements. Similarly to the Sec. 4.3, we also validate (with the use of FLEXPART) the estimated source terms with the IAEA reported measurements and compute associated R value for each scatter plot in Fig. 11. The R value is slightly better for the FAST dataset (0.710) than for the RAW dataset (0.695) while it is 0.578 for the RAW dataset and even lower for the WEEKS dataset (0.288). These results support the hypothesis that better temporal resolution of measurements are beneficial for source term inversion.

## 5   Conclusions

We have investigated the occurrence of $^{106}$Ru in Europe in the fall of 2017. We have used data from the Czech monitoring network which also includes measurement data from novel real-time monitoring systems. Based on this case study, it can be concluded that both systems are suitable for the task of rapid detection of radioactive contamination in the atmosphere at the level of mBq/m$^3$. Each of the developed devices employs a different sampling/measurement procedure and therefore there are also different possibilities of their integration into a large-scale monitoring network. The combination of the AMARA system and laboratory measurement seems to be an optimal setup balancing response sensitivity and timeliness. On the other hand, the CEGAM system can be operated unattended in remote locations in a stand-by regime with a relatively low power consumption and can be switched to emergency regime if needed. Regarding the employed electrically cooled HPGe detectors, they proved to be resilient enough to be deployed long term. For the past three years we have not experienced any malfunction or need of excessive maintenance so the only drawback of HPGe detectors are the accompanied costs compared to the NaI(Tl) setup which we used in the past.

Using the inversion modeling technique, we have compared the results obtained from four datasets ranging from raw data, using the standard measuring procedure, to real-time monitoring data with a much better temporal resolution. The results have

been compared with the published state-of-the-art estimates of the $^{106}$Ru release in 2017. Based on this comparison, we have observed that the results obtained using real-time monitoring data are comparable in terms of the total estimated release and are better for the temporal specification of the release, while they are consistent with the previously reported findings regarding the location of the $^{106}$Ru source term.

In addition, we have compared our results based on the Czech monitoring data with the dataset reported by the IAEA, which has a much better spatial coverage. The source location results have been compared using NMSE, NNR, FMS, and FB coefficients between the IAEA results and the results based on the Czech monitoring data. We have concluded that the real-time monitoring data result is close to the IAEA result. Four source location hypotheses have been tested based on the correlation coefficient between the IAEA measurements and the model reconstruction using Czech monitoring data. Here, the results are in agreement with previous studies, with the Mayak location being the most probable ($R = 0.381$) in comparison with Dimitrovgrad ($R = 0.349$), southern Romania ($R = 0.139$), and the location to the north of Mayak ($R = 0.109$).

Concerning the real time monitoring capabilities of the Czech radiation monitoring network, we have shown that a single operating device can enhance the inverse modeling predictions even for a relatively low radionuclide concentration at the level of mBq/m$^3$. Although the continental scale scenario such as the $^{106}$Ru case in the 2017 may not be ideal for quantification of a real-time monitoring system benefits due to the diffusion over several days transport, we believe that the benefits are still observable. It is safe to state that the installation of multiple devices such as AMARA and CEGAM over a larger region (on a European scale) would certainly yield additional improvements in source location and in source term estimation in the event of a radionuclide atmospheric release.

*Code and data availability.* The Czech datasets are freely available as the supplement of this paper. The HYSPLIT and FLEXPART models are open source and are freely available from their developers. Reference MATLAB implementations of algorithms can be obtained from the corresponding author upon request.

*Author contributions.* OT designed and performed the experiments and wrote the paper. MH designed experiments, conducted measurements, and wrote the paper. NE performed FLEXPART simulations and commented on the manuscript. VS commented on the manuscript and inversion procedure.

*Competing interests.* The authors declare that they have no conflict of interest.

*Acknowledgements.* This work was supported by the Czech Science Foundation, grant no. GA20-27939S, and by the VI20152018042 project funded by the Ministry of the Interior of the Czech Republic.

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
