# Peer review of "Real-time measurement of radionuclide concentrations and its impact on inverse modeling of 106Ru release in the fall of 2017"

_Atmospheric Measurement Techniques, 2020_

## Referee Comment (RC1) · Anonymous Referee #2 · 9 Sep 2020

The authors report a new real-time measurement of radionuclides in airborne particles that yields a better temporal resolution of low concentration radionuclide measurement. This improved temporal resolution in turn improves the reliability of the inverse modeling of the Ru-106 event in fall 2017, during a period when the new device was already employed. The authors conclude that if applied in multiple locations across Europe, the possible location of a radionuclide release may be identified and its source term estimated more quickly as the inverse modeling is more reliable. Previous experiences have shown that the modeling of radionuclide transport and dispersion in the atmosphere is often complicated by the low temporal resolution of the measurements. However long sampling periods and spectrum acquisition times are required in order

to quantify low activity concentrations hundreds or thousands of kilometers away from the location of the release. The authors present a procedure that delivers spectra for analysis during air sampling, thus allowing the retrieval of the temporal evolution during the sampling period. The modeling part of the study confirms that the improved temporal resolution renders the inverse modeling more accurate. I agree that better temporal resolutions of radionuclide concentrations at very low concentrations are highly desirable for environmental radioactivity monitoring and the authors present a new and promising approach in this direction. I thus support the publication of this study in AMT. However I recommend the following revisions before publication:

Abstract: I suggest that the capabilities of the new device are specified more clearly in the abstract, e.g.: p.1 Line 7: . . . gamma-ray counting of aerosol filters and allow us to determine the moment when Ru-106 arrived at the measurement site within XXX minutes and activity concentrations as low as XXX can be detected in 4-hour intervals.

Section 2: Measurement methodology and datasets The descriptions of the new AMARA and CEGAM systems are very short. A reader of "Atmospheric Measurement Techniques" might be interested in a some details about the techniques which are omitted in the manuscript. I recommend revising Sections 2.2.1 and 2.2.2 such that they at minimum answer the following questions: 1) What is the efficiency of the HPGe-detector relative to the 3"-NaI 2) How is the detector cooled (electrically or liquid nitrogen) 3) How stable is the temperature of the Germanium crystal on a hot summer day or a cold winter day? 4) Do variations in relative humidity affect the detector? 5) Is the energy calibration affected by variations in ambient conditions (temperature or humidity) ? Is there an automatic recalibration procedure, e.g. with a reference peak? 6) How was the efficiency calibration performed? 7) Was True Coincidence Summation (TCS) considered as it is for the standard sampling and measurement procedure? 8) What is the interval between the spectra in the case of AMARA? Signal Treatment: 1) How accurately can the time of arrival be determined (see my suggestion "within XXX minutes" for the abstract) 2) How was the AMARA reconstruction (black line in

Figure 2B) performed? What are the corresponding time intervals and uncertainties? Are uncertainties of one interval affected by the deposition of Ru-106 during previous intervals? Further, it is obvious from figure 2 that the plume continued for longer than is on display here. Why is the remaining part not shown? Was it also split into 4-hour CEGAM intervals for inverse modeling runs?

Section 5: Conclusions This section almost reads as if the authors have carried out a pure modeling study. In its current form it does little justice to the technical progress that they achieved and which justifies publication in a technical journal. I suggest that the authors provide a brief summary of the new measurement device and its advantages in this section.

Other minor issues: p. 1, Line 24: "several hundred TBq": This needs one or more references where the source term is estimated. p. 2, Line 9: I suggest rewriting this sentence along the lines of: "Since medical sources and RTG would neither explain the occurrence of Ru-106 nor the large source of several hundred TBq, fresh nuclear fuel is the most likely candidate. " P4. Figure 2 (A): The y-scale is missing. The reader needs to know how many keVs are displayed. P.5 Line 7: (Hyza and Rulik, 2017) should be Hyza and Rulik (2017) P.5 Line 9: it should be mentioned that the MDAC worsens for one particular interval if some Ru-106 was already deposited in a preceding interval p.6 Line 27: I propose "location of the release" instead of "location of the source of the release" p.9 Line 11: (Tichy et al., 2016) should be Tichy et al. (2016) p.10 Line 28: I suggest "During the period in question" instead of "In the assumed period" p.12 Line 10: I suggest "A release at location 4 in southern Romania would contradict ground-based observations to the east of the location was thus also rejected (see Masson et al., 2019)." p.12 Line 16: The colour code is already described in the caption of Figure 6 and can be omitted here.

---

## Referee Comment (RC2) · Anonymous Referee #1 · 17 Sep 2020

Review of "Real-time measurement of radionuclide concentrations and its impact on inverse modeling of Ru-106 release in the fall of 2017"

General comments

This paper presents an inverse modelling study of airborne Ru-106 detections made in the Czech Republic in the fall of 2017. An existing inverse modeling algorithm, already successfully applied in earlier studies, is used and the results are compared (and found compatible) with other studies that considered the Ru-106 detections. Furthermore, in this study the authors also describe two new detection systems. The motivation for the new detection systems is that they will provide observations with a higher temporal

resolution, which is obtained by reducing the sampling time (CEGAM system) or by measuring during sampling (AMARA system). Finally, the authors consider different data sets to perform the inverse modelling. They conclude that inverse modelling using data with shorter sampling times (thus having a higher temporal resolution) performs equally well or better than inverse modelling using data having sampling times of a few days up to one week. This paper is relevant and results are compatible with previous studies, although I think the conclusions related to the added value of the new measurement systems are not well supported by the results. Furthermore, the chosen case study – although being a very important and interesting case – is likely not well-suited to fully demonstrate the added value of such systems given the large geotemporal scales of the Ru-106 release (having source-receptor distances of thousands of kilometers).

Specific comments

In the abstract, the authors wrote: "Since reasonable temporal resolution of concentration measurements is crucial for proper source term reconstruction, the standard one week sampling interval could be limiting". Although it is sensible that better temporal resolution will lead to better source reconstruction, I'm wondering how important the limiting effect is. The effect is likely case-dependent, and in particular more pronounced for problems with shorter geotemporal scales. In that light, the Ru-106 case might not fully demonstrate the added value of short sampling times. A test with a fictitious source and fictitious measurements would be instructive (one test at scales of a few hunderds of kilometers, and another test at a few thousands of kilometers). The fictitious experiment could demonstrate and quantify the limiting effect of long sampling times in a more controlled way.

The AMARA and CEGAM measurement system descriptions are not clear to me. Specifically: p 5, line 9: "The achieved MDAC for Ru-106 is at a level of 1 mBq/m3 per one-hour integration time and 12 hours of sampling." Does this mean that an activity concentration measurement is available every hour, and that the filter is renewed every 12 hours? And for the CEGAM system, a measurement is available every 4

hours, and the filter is renewed every 4 hours? What is the philosophy of having two different systems, and will both systems be used and maintained in the coming years? p 5, line 8: the reference to Fig 2 is slightly confusing since no 4-hour averaging is applied for the AMARA system?

p 6, line 5: "Unfortunately, the CEGAM system was not yet operational during the Ru-106 incident but we have simulated its output by integrating the AMARA results in a 4 hours window." Some additional information would be helpful here. If CEGAM pseudo-observations are used based on the AMARA observations, then they would contain the same information? I assume the simulated output is not used for the inversion, but it would be good to confirm this in the text.

Section 2.3: Dataset description: I think it would be good to add a figure or table that summarizes the different datasets (range of observed activity concentrations, number of observations, number of (non-)detections. After consulting the Supplementary Information, I am a bit worried that the differences between data set "RAW" and "FAST" are too small to be significant. Also, why is the integration window set to be between 3 and 13 hours? From Sections 2.2.1 and 2.2.2, I expected that measurements from the AMARA system would be available every hour (and measurements from the CEGAM system every 4 hours)?

p 9 line 5: it would be instructive to get an estimate of the values used for \sigma_{length}^2 in the calculation of the inverse covariance matrix R.

Section 4.1: Atmospheric transport modeling: Numerical weather prediction data, which is used to drive the atmospheric transport model Hysplit, was available every 6 hours. This is likely sufficient for the geotemporal scales of the problem. However, it might not if one wants to explore the added value of measurements with sub-daily sampling periods.

Table 3: Can the authors think of any reason why the release length is significantly different for the four considered locations when using the data sets "WEEKS" and "CUT",

but not when using the datasets "RAW" and "FAST"? If one does not assume a priori that a short release period is better, Table 3 could be interpreted as if data set "FAST" gives less information regarding the release duration than "WEEKS", as it is less sensitive to the location. Also, I guess that the regularization will have a larger impact on the release duration than the choice of the data set. From these considerations, I am not convinced that the real-time monitoring data results in a better temporal specification of the release, as stated in Conclusions.

Figure 7 is important for assessing the quality of the inverse modeling results that were obtained using different data sets, by comparing simulated activity concentrations with the IAEA measurements. However, Figure 7 seems to suggest that the temporal resolution of the observations do not really matter for this case. Perhaps other metrics might reveal an improvement from the use of higher temporal resolution, but I doubt that that will be the case for this specific case study (large geotemporal scales and 6-h meteorological data). In the same Figure 7, data set "RAW" performs slightly worse than data set "WEEKS". Do the authors have an explanation for that? From temporal resolution considerations, I would expect that "FAST" performs equally well or better than "RAW", and "RAW" equally well or better than "WEEKS". Also, from Figure 4 and knowing the true source location, I do not see why the results using the "FAST" data set would be better than the results from the other data sets. Concerning the table in Figure 9, I wonder whether other metrics would come to different conclusions (the NMSE, although widely used, is not unbiased, see Poli and Cirillo, 1993). These considerations make it hard for me to agree with the statement made on p 17 line 5.

p 17 line 1: how are the probabilities of the source location calculated? Is it the evidence / marginal likelihood, but normalized so that its sum over the whole domain equals 1?

In Conclusions, the authors wrote: "It is safe to state that the installation of multiple devices such as AMARA and CEGAM over a larger region (ona European scale) would certainly yield additional improvements in source location and in source term estimation in the eventof a radionuclide atmospheric release." There is a trade-off between detector sensitivity and the sampling length (more observations will have a higher minimum detectable concentration). I suggest to briefly discuss this trade-off also in the conclusions. Also, although I agree that there is potential in using observations with higher temporal resolution, I don't think that its added value is clearly demonstrated in this study.

Technical corrections

p 3, line 1: location –> localisation p 10, line 13: ". . . and run for the period . . ." –> ". . . and release particles during the period ..." p 12, line 16: "The estimated source terms are displayed for the RAW dataset using blue lines, for the WEEKS dataset using magenta lines, for the FAST dataset using red lines, and for the CUT dataset using green lines." –> I suggest to omit this sentence at this is already mentioned in the caption of Figure 6.

References:

Poli, A. A., & Cirillo, M. C. (1993). On the use of the normalized mean square error in evaluating dispersion model performance. Atmospheric Environment. Part A. General Topics, 27(15), 2427-2434.

---

## Author Comment (AC1) · 1 Dec 2020

We would like to thank you for providing us a detailed review of our manuscript. We are glad that we can submit a revision of our paper. In the following text, we will respond to all comments.

**The authors report a new real-time measurement of radionuclides in airborne particles that yields a better temporal resolution of low concentration radionuclide measurement. This improved temporal resolution in turn improves the reliability of the inverse modeling of the Ru-106 event in fall 2017, during a period**

[Figure]

when the new device was already employed. The authors conclude that if applied in multiple locations across Europe, the possible location of a radionuclide release may be identified and its source term estimated more quickly as the inverse modeling is more reliable. Previous experiences have shown that the modeling of radionuclide transport and dispersion in the atmosphere is often complicated by the low temporal resolution of the measurements. However long sampling periods and spectrum acquisition times are required in order to quantify low activity concentrations hundreds or thousands of kilometers away from the location of the release. The authors present a procedure that delivers spectra for analysis during air sampling, thus allowing the retrieval of the temporal evolution during the sampling period. The modeling part of the study confirms that the improved temporal resolution renders the inverse modeling more accurate. I agree that better temporal resolutions of radionuclide concentrations at very low concentrations are highly desirable for environmental radioactivity monitoring and the authors present a new and promising approach in this direction. I thus support the publication of this study in AMT. However, I recommend the following revisions before publication:

**Specific comments:**

**Abstract:** I suggest that the capabilities of the new device are specified more clearly in the abstract, e.g.: p.1 Line 7: . . . gamma-ray counting of aerosol filters and allow us to determine the moment when Ru-106 arrived at the measurement site within XXX minutes and activity concentrations as low as XXX can be detected in 4-hour intervals.

*Authors response*: Thank you for this suggestion, we have modified the abstract accordingly.

*Changes made in the paper*:  We added the sentence to the abstract.

**Section 2: Measurement methodology and datasets The descriptions of the new AMARA and CEGAM systems are very short. A reader of "Atmospheric Measurement Techniques" might be interested in a some details about the techniques which are omitted in the manuscript. I recommend revising Sections 2.2.1 and 2.2.2 such that they at minimum answer the following questions: 1) What is the efficiency of the HPGe-detector relative to the 3"-NaI 2) How is the detector cooled (electrically or liquid nitrogen) 3) How stable is the temperature of the Germanium crystal on a hot summer day or a cold winter day? 4) Do variations in relative humidity affect the detector? 5) Is the energy calibration affected by variations in ambient conditions (temperature or humidity) ? Is there an automatic recalibration procedure, e.g. with a reference peak? 6) How was the efficiency calibration performed? 7) Was True Coincidence Summation (TCS) considered as it is for the standard sampling and measurement procedure? 8) What is the interval between the spectra in the case of AMARA? Signal Treatment: 1) How accurately can the time of arrival be determined (see my suggestion "within XXX minutes" for the abstract) 2) How was the AMARA reconstruction (black line in Figure 2B) performed? What are the corresponding time intervals and uncertainties? Are uncertainties of one interval affected by the deposition of Ru-106 during previous intervals? Further, it is obvious from figure 2 that the plume continued for longer than is on display here. Why is the remaining part not shown? Was it also split into 4-hour CEGAM intervals for inverse modeling runs?**

*Authors response*:  We acknowledged that more details on measuring systems pre-
    sented in the paper are need.    We extended the description of the
    AMARA/CEGAM systems significantly and with respect to the reviewer comment.
    All important parameters are now summarized in Table 1 in the paper. We also
    added discussion on signal processing.

The activity uncertainty is indeed affected by the previous deposition, we have included this into a text together with mentioning also the influence of radon background.

Regarding the displayed data - the end on the time axis corresponds to the filter change. We have included this info in the figure caption. Consequent samples were taken with a higher frequency and the ruthenium activity decreased, therefore the real time measurement provided less and less useful information. 4-hours intervals were computed only for the first sampling interval capturing the arrival of contamination.

*Changes made in the paper*: Section 2 was extended on Section 2.3 to provide all above mentioned information. Also, the table with important parameters on AMARA and CEGAM systems was added to the paper. Caption of Fig. 2 was expanded to provide further information.

**Section 5: Conclusions This section almost reads as if the authors have carried out a pure modeling study. In its current form it does little justice to the technical progress that they achieved and which justifies publication in a technical journal. I suggest that the authors provide a brief summary of the new measurement device and its advantages in this section.**

*Authors response*: Indeed, we did not intend to focus mostly on modeling in conclusion, which we, however, did. Thank you, we extended significantly the first paragraph to give more attention to the introduced modeling systems.

*Changes made in the paper*: The first paragraph of the conclusion is extended significantly.

**Minor issues:**

**p. 1, Line 24: "several hundred TBq": This needs one or more references where the source term is estimated.**

*Authors response*: Although these findings are referenced later, we agree that they should be referenced also here and we added relevant references.

**p. 2, Line 9: I suggest rewriting this sentence along the lines of: "Since medical sources and RTG would neither explain the occurrence of Ru-106 nor the large source of several hundred TBq, fresh nuclear fuel is the most likely candidate. "**

*Authors response*: The sentence was rewritten accordingly.

**P4. Figure 2 (A): The y-scale is missing. The reader needs to know how many keVs are displayed.**

*Authors response*: Agree. The ROI width was specified in the figure caption.

**P.5 Line 7: (Hyza and Rulik, 2017) should be Hyza and Rulik (2017)**

*Authors response*: Corrected.

**P.5 Line 9: it should be mentioned that the MDAC worsens for one particular interval if some Ru-106 was already deposited in a preceding interval**

*Authors response*: Agree. The brief discussion of possible influences on MDAC are mentioned in newly extended section 2.

**p.6 Line 27: I propose "location of the release" instead of "location of the source of the release"**

*Authors response*: Thank you, we agree.

**p.9 Line 11: (Tichy et al., 2016) should be Tichy et al. (2016)**

*Authors response*: Corrected.

**p.10 Line 28: I suggest "During the period in question" instead of "In the assumed period"**

*Authors response*: We agree with this suggestion.

**p.12 Line 10: I suggest "A release at location 4 in southern Romania would contradict ground-based observations to the east of the location was thus also rejected (see Masson et al., 2019)."**

*Authors response*: We reformulated this accordingly.

**p.12 Line 16: The colour code is already described in the caption of Figure 6 and can be omitted here.**

*Authors response*: We removed the color code description from here.

---

## Author Comment (AC2) · 1 Dec 2020

We would like to thank you for providing us a detailed review of our manuscript. We are glad that we can submit a revision of our paper. In the following text, we will respond to all comments.

**This paper presents an inverse modelling study of airborne Ru-106 detections made in the Czech Republic in the fall of 2017. An existing inverse modeling algorithm, already successfully applied in earlier studies, is used and the results are compared (and found compatible) with other studies that considered**

[Figure]

the Ru-106 detections. Furthermore, in this study the authors also describe two new detection systems. The motivation for the new detection systems is that they will provide observations with a higher temporal resolution, which is obtained by reducing the sampling time (CEGAM system) or by measuring during sampling (AMARA system). Finally, the authors consider different data sets to perform the inverse modelling. They conclude that inverse modelling using data with shorter sampling times (thus having a higher temporal resolution) performs equally well or better than inverse modelling using data having sampling times of a few days up to one week. This paper is relevant and results are compatible with previous studies, although I think the conclusions related to the added value of the new measurement systems are not well supported by the results. Furthermore, the chosen case study – although being a very important and interesting case – is likely not well-suited to fully demonstrate the added value of such systems given the large geotemporal scales of the Ru-106 release (having source-receptor distances of thousands of kilometers).

**Specific comments:**

In the abstract, the authors wrote: "Since reasonable temporal resolution of concentration measurements is crucial for proper source term reconstruction, the standard one week sampling interval could be limiting". Although it is sensible that better temporal resolution will lead to better source reconstruction, I'm wondering how important the limiting effect is. The effect is likely case-dependent, and in particular more pronounced for problems with shorter geotemporal scales. In that light, the Ru-106 case might not fully demonstrate the added value of short sampling times. A test with a fictitious source and fictitious measurements would be instructive (one test at scales of a few hunderds of kilometers, and another test at a few thousands of kilometers). The fictitious

[Figure]

**experiment could demonstrate and quantify the limiting effect of long sampling times in a more controlled way.**

*Authors response***:** We agree with the reviewer that the Ru-106 case is not a perfect match to study the influence of a fast measuring system in details. We prefer to avoid a synthetic study since its results would be sensitive to our simulation setup. The Ru-106 event was the first significant release with a fully operational AMARA system and it is also well studied in the literature which allows discussion of the obtained results. To provide more solid evidence on the added value of the fast measurements, we extended the paper by additional simulation using the FLEXPART atmospheric transport model (see details in other responses below). The new simulation using a completely different simulation tool resulted in the same conclusion, supporting our previous claim that was based on a single model (HYSPLIT) and thus could have been obtained by chance.

*Changes made in the paper***:** We employ the FLEXPART model to the same datasets to demonstrate that the results are not obtained by change but are systematic.

**The AMARA and CEGAM measurement system descriptions are not clear to me. Specifically: p 5, line 9: "The achieved MDAC for Ru-106 is at a level of 1 mBq/m3 per one-hour integration time and 12 hours of sampling." Does this mean that an activity concentration measurement is available every hour, and that the filter is renewed every 12 hours? And for the CEGAM system, a measurement is available every 4 hours, and the filter is renewed every 4 hours? What is the philosophy of having two different systems, and will both systems be used and maintained in the coming years? p 5, line 8: the reference to Fig 2 is slightly confusing since no 4-hour averaging is applied for the AMARA system?**

*Authors response***:** There are two time intervals affecting the final MDAC - the duration of measurement and the duration of the sampling. Spectra in the AMARA system are measured (sampling duration) every 5 minutes and therefore arbitrary sums (measurement duration) could be computed afterwards. On the other hand, CEGAM system is limited by the time step of carousel mechanism therefore the sampling duration is equal to measurement duration.

*Changes made in the paper*: We have extended section 2 in order to better describe the measurement/sampling logistics. The difference between both systems and their intended use is also briefly discussed.

**p 6, line 5: "Unfortunately, the CEGAM system was not yet operational during the Ru-106 incident but we have simulated its output by integrating the AMARA results in a 4 hours window." Some additional information would be helpful here. If CEGAM pseudo-observations are used based on the AMARA observations, then they would contain the same information? I assume the simulated output is not used for the inversion, but it would be good to confirm this in the text.**

*Authors response*: Indeed, the use of simulated output would be pointless since it would contained the same information. All data used for inversion are based on AMARA system, the CEGAM system was set to operational regime later. We agree that the sentence was not clear and we state clearly this fact in the current version of the manuscript.

*Changes made in the paper*: We reformulated the sentence to avoid misunderstanding.

**Section 2.3: Dataset description: I think it would be good to add a figure or table that summarizes the different datasets (range of observed activity concentrations, number of observations, number of (non-)detections. After consulting the Supplementary Information, I am a bit worried that the differences between data set "RAW" and "FAST" are too small to be significant. Also, why is the integration window set to be between 3 and 13 hours? From Sections 2.2.1 and 2.2.2, I**

**expected that measurements from the AMARA system would be available every hour (and measurements from the CEGAM system every 4 hours)?**

*Authors response*: Thank you for pointing this out, we agree that graphical representation of measurements would be instructive. Since the main differences can be observed in the case of the Prague station (equipped with the AMARA system), we provide a figure that summarizes measurements from this station.

The integration window was set adaptively to maintain the sufficient response to the Ru-106 activity. Difference between the real-time measurement values and values obtained by the measurement of the whole filter in laboratory was within approx. 15 %. This error margin is also compatible with our previous findings where we compared the laboratory values and real-time values of natural Be-7.

*Changes made in the paper*: We added a figure with visualization of measurements from Prague as well as related description in the text.

**p 9 line 5: it would be instructive to get an estimate of the values used for $\sigma_{length}^2$ in the calculation of the inverse covariance matrix R.**

*Authors response*: Indeed, we miss out to define the $\sigma_{length}$ coefficient in the text which is now corrected. It is defined as $\sigma_{length} = \frac{measurement\,hours}{6}$ where the 6 hours window is motivated by the GFS data resolution. Varying the length of this window does not affect the results significantly.

*Changes made in the paper*: We define this coefficient in Sec. 3.2 in the revised manuscript.

**Section 4.1: Atmospheric transport modeling: Numerical weather prediction data, which is used to drive the atmospheric transport model Hysplit, was available every 6 hours. This is likely sufficient for the geotemporal scales of the**

**problem. However, it might not if one wants to explore the added value of measurements with sub-daily sampling periods.**

*Authors response*: We agree that the 6-hours resolution of the meteorology may be limiting and may somehow blur the results. Therefore, we run a new simulation using the FLEXPART model driven by 3-hour meteorological analyses from the European Centre for Medium-Range Weather Forecasts (ECMWF) and, subsequently, we select 3 hours temporal resolution of the output grid. We conclude from the estimated source terms that better temporal resolution of measurements improves the temporal specificity of the source term. We demonstrate this in the case of source term estimation from the most probable location 2, Mayak, in Sec. 4.4.

*Changes made in the paper*: We extend the paper by FLEXPART simulation with higher temporal resolution. The FLEXPART configuration is given in Sec. 4.1.2 while the results for the most probable location, Mayak, are given in Sec. 4.4.

**Table 3: Can the authors think of any reason why the release length is significantly different for the four considered locations when using the data sets "WEEKS" and "CUT", but not when using the datasets "RAW" and "FAST"? If one does not assume a priori that a short release period is better, Table 3 could be interpreted as if data set "FAST" gives less information regarding the release duration than "WEEKS", as it is less sensitive to the location. Also, I guess that the regularization will have a larger impact on the release duration than the choice of the data set. From these considerations, I am not convinced that the real-time monitoring data results in a better temporal specification of the release, as stated in Conclusions.**

*Authors response*: The LS-APC algorithm was designed to minimize the number of tuning parameters (they are estimated from the data) leaving its result sensitive only

to the initial conditions. As demonstrated on the ETEX dataset in the original publication, it is rather insensitive even to the initial conditions. Further confirmation can be found recently in (Tichý, O., Ulrych, L., Šmídl, V., Evangeliou, N., and Stohl, A.: On the tuning of atmospheric inverse methods: comparisons with the European Tracer Experiment (ETEX) and Chernobyl datasets using the atmospheric transport model FLEXPART, Geosci. Model Dev., 13, 5917–5934, https://doi.org/10.5194/gmd-13-5917-2020, 2020.).

In fact, LS-APC assigns a higher prior probability to shorter releases than to longer ones. The preference is rather weak and informative data overrule this prior. However, this is probably the reason for different lengths e.g. in the case of the WEEKS dataset. When the observations could be explained by a shorter release, LS-APC considers it a more likely solution.

*Changes made in the paper*: We extended discussion of the results in Sec. 4.2 and also add a reference to the sensitivity study of the used LS-APC algorithm to Sec. 3.1.

**Figure 7 is important for assessing the quality of the inverse modeling results that were obtained using different data sets, by comparing simulated activity concentrations with the IAEA measurements. However, Figure 7 seems to suggest that the temporal resolution of the observations do not really matter for this case. Perhaps other metrics might reveal an improvement from the use of higher temporal resolution, but I doubt that that will be the case for this specific case study (large geotemporal scales and 6-h meteorological data). In the same Figure 7, data set "RAW" performs slightly worse than data set "WEEKS". Do the authors have an explanation for that? From temporal resolution considerations, I would expect that "FAST" performs equally well or better than "RAW", and "RAW" equally well or better than "WEEKS". Also, from Figure 4 and knowing the true source location, I do not see why the results using the "FAST" data set would be better than the results from the other data sets. Concerning the**

**table in Figure 9, I wonder whether other metrics would come to different conclusions (the NMSE, although widely used, is not unbiased, see Poli and Cirillo, 1993 - Poli, A. A., & Cirillo, M. C. (1993). On the use of the normalized mean square error in evaluating dispersion model performance. Atmospheric Environment. Part A. General Topics, 27(15), 2427-2434.). These considerations make it hard for me to agree with the statement made on p 17 line 5.**

*Authors response***:** We agree that the agreement with the IAEA measurements on former Figure 7 is rather insensitive to the choice of the temporal resolution. Therefore, we perform an additional simulation using the FLEXPART model with 3-hour temporal resolution and present these results in the updated version of the manuscript. The results from the FLEXPART runs are summarized in Sec. 4.4 for location 2, Mayak. We believe that the temporal specificity of the FAST dataset is better demonstrated there. Although the FLEXPART slightly overpredicted some of the IAEA observation, the estimation using the FAST dataset provides the best fit.

We are grateful to the reviewer for pointing out deficiencies of the NMSE coefficient. In the current version of the manuscript, we use four coefficients: the normalized mean square error (NMSE), the normalized mean square error of the distribution of the normalized ratios (NNR) suggested by Poli and Cirillo, and also other coefficients: the figure of merit in space (FMS) and the fractional bias (FB). In all cases, the results by the FAST dataset are the closest to the IAEA result.

*Changes made in the paper***:** First, we extended the manuscript by the FLEXPART simulation with finer temporal resolution and study the results for location 2, Mayak, in Sec. 4.4. Second, we extended Sec. 4.3 by additional coefficients, NNR, FMS, and FB.

**p 17 line 1: how are the probabilities of the source location calculated? Is it the**

**evidence / marginal likelihood, but normalized so that its sum over the whole domain equals 1?**

*Authors response***:** The marginal log-likelihood is normalized using the maximum of each domain, hence, the maximum of each normalized domain is equal to 1. This information was missing in the manuscript and we added it to the present version.

*Changes made in the paper***:** The information on the normalization of the displayed marginal likelihood is added to Sec. 3.3.

**In Conclusions, the authors wrote: "It is safe to state that the installation of multiple devices such as AMARA and CEGAM over a larger region (on European scale) would certainly yield additional improvements in source location and in source term estimation in the event of a radionuclide atmospheric release." There is a trade-off between detector sensitivity and the sampling length (more observations will have a higher minimum detectable concentration). I suggest to briefly discuss this trade-off also in the conclusions. Also, although I agree that there is potential in using observations with higher temporal resolution, I don't think that its added value is clearly demonstrated in this study.**

*Authors response***:** The trade-off mentioned by the reviewer is now discussed in section 2, we acknowledged the reviewers' remark. We also mentioned in conclusion that there is possible limitation of the continental scale scenario, however, we believe that the effect of real-time monitoring system is still observable.

*Changes made in the paper***:** We extended section 2 significantly and we also discuss some issues regarding the scale of the experiment in conclusion.

**Minor issues:**

**p 3, line 1: location –> localisation**

*Authors response*: Thank you, we corrected this typo.

**p 10, line 13: ". . . and run for the period . . ." –>". . . and release particles during the period ..."**

*Authors response*: We reformulated this accordingly.

**p 12, line 16: "The estimated source terms are displayed for the RAW dataset using blue lines, for the WEEKS dataset using magenta lines, for the FAST dataset using red lines, and for the CUT dataset using green lines." –> I suggest to omit this sentence as this is already mentioned in the caption of Figure 6.**

*Authors response*: Indeed, we removed the color code description from here.

---

## Author Response (AR2)

**Response to the Reviews**

We would like to thank to both reviewers for valuable comments. We are glad that we can submit technical corrections of our paper. In the following text, we will respond to all comments.

**Comments from referee #1:**

→ I would like to acknowledge the work that the authors have done in revising the manuscript.

However, I'm still a bit sceptical about the conclusion that changing the measurements for the Prague station would result in an improved inversion. This is mainly because only the correlation is used in Figs 8, 9 and 11 (as I mentioned in my first review, concerning Fig 7 of the original manuscript). The scatterplots in Fig 11 look all very similar, apart from a few outliers, which dominate the correlation. I would advice other metrics than the correlation, such as: the correlation of log(concentration), NMSE, FB, NNR, WNNR, ...

**Authors response**: We agree with the reviewer that the scatters (especially for Fig. 11) are quite similar and outliers may dominate. Therefore, we compute the correlation coefficients of logarithm of concentrations (Rlog) as suggested by the reviewer for all scatter plots in the paper. Note that the Rlog coefficients are in agreement with R coefficients; hence, this support our conclusions.

→ **Technical comments**:
- the units for sigma_length are not compatible with Eq 10
- l 11, p 15: tothe --> to the

**Authors response**: Thank you, we corrected both issues.

[revised manuscript text omitted]
} = \mathrm{diag}\sqrt{\boldsymbol{\sigma}_{\mathrm{abs}}^2 + \left(\boldsymbol{\sigma}_{\mathrm{rel}} \circ \mathbf{y}\right)^2 + \frac{1}{\boldsymbol{\sigma}_{\mathrm{length}}^2}}, \tag{10}$$

where $\boldsymbol{\sigma}_{\mathrm{abs}}^2$ is the absolute measurement error which is selected between 0.2 and 1.4 mBq based on the maximum a posteriori estimate, $\boldsymbol{\sigma}_{\mathrm{rel}}$ is the uncertainty level of measurements, which is between 5.5 and 30% for our dataset, and $1/\boldsymbol{\sigma}_{\mathrm{length}}^2$ is the term considering the length of the measurement as $\boldsymbol{\sigma}_{\mathrm{length}} = \frac{\text{measurement hours}}{6 \text{ hours}}$ (mBq) where the selection of 6 hours window is motivated by the GFS meteorological data resolution. Here, a shorter measurement time implies higher uncertainty and a longer measurement time implies lower uncertainty.

**3.3 Variational Bayes inference and source location**

Within the variational Bayes (VB) framework (Šmídl and Quinn, 2006), the posterior distributions are found in the same functional form as their priors. The moments of the posteriors are determined using an iterative algorithm with details in Tichý et al. (2016). Here, the reference Matlab implementation can be downloaded as a supplement. The method will be denoted here as the LS-APC-VB method.

Moreover, we consider the scenario where we have a finite set of SRS matrices $\{\mathbf{M}_1, \mathbf{M}_2, \ldots, \mathbf{M}_r\}$, representing different considered locations of the release here. For each SRS matrix from the set, we can evaluate the posterior probability $p\left(\mathbf{M} = \mathbf{M}_k | \mathbf{y}\right)$ as

[revised manuscript text omitted]